# Eribulin Mesylate Improves Cisplatin-Induced Cytotoxicity of Triple-Negative Breast Cancer by Extracellular Signal-Regulated Kinase 1/2 Activation

**DOI:** 10.3390/medicina58040547

**Published:** 2022-04-15

**Authors:** Hyemi Ko, Myungsun Lee, Eunyoung Cha, Jiyoung Sul, Junbeom Park, Jinsun Lee

**Affiliations:** 1Department of Surgery, Chungnam National University Hospital, Jung-gu, Daejeon 35015, Korea; kohyemi86@gmail.com (H.K.); jysul@cnuh.co.kr (J.S.); jbpark@cnuh.co.kr (J.P.); 2Biomedical Research Institute, Chungnam National University Hospital, Jung-gu, Daejeon 35015, Korea; myunglee@cnu.ac.kr (M.L.); skybird457@hanmail.net (E.C.); 3Department of Surgery and Research Institute for Medicinal Sciences, Chungnam National University College of Medicine, Jung-gu, Daejeon 35015, Korea

**Keywords:** eribulin, cisplatin, triple-negative breast cancer, ERK

## Abstract

*Background and Objectives*; Triple-negative breast cancer (TNBC) is associated with poor patient prognosis because of its multiple molecular features. Thus, more effective treatment for TNBC is urgently needed. This study determined the possible involvement of ERK1/2 activation in cisplatin-induced cytotoxicity in TNBC by providing additional eribulin treatment. *Materials and Methods*; We investigated cell viability and apoptosis caused by eribulin, cisplatin, or co-treatment in HCC38, MDA-MB-231, and SKBR3 human breast cancer cells. *Results*; Cisplatin significantly lowered cell viability and caused high apoptotic cell death in all breast cancer cell lines. The viability of TNBC cells was significantly lower in the group co-treated with cisplatin and eribulin than in the cisplatin-only treatment group. Additional eribulin treatment significantly enhanced PARP cleavage and caspase-3 activity in cisplatin-treated TNBC cells. Moreover, cisplatin treatment activated ERK1/2 in all breast cancer cell lines. The cisplatin and eribulin combination synergistically activated ERK1/2 in TNBC cells compared with the cisplatin-only treatment. Administration of the ERK1/2 inhibitor PD98059 increased the viability of TNBC cells treated with cisplatin plus eribulin. *Conclusions*; Eribulin could synergize the cytotoxic and apoptotic activities of cisplatin and increase ERK1/2 activation, thus enhancing anti-cancer effects against TNBC cells.

## 1. Introduction

As a clinically challenging type of cancer, triple-negative breast cancer (TNBC) is found more often in younger women (<50 years) [1,2,3,4] and African American women who are premenopausal [4,5,6]. In addition, BRCA1(Breast cancer type 1) mutation carriers [7,8] are more likely to have TNBC, and TNBC is associated with more aggressive tendencies compared with other subtypes [1,9].

Chemotherapy still represents the main treatment in both the initial and metastatic stages of TNBC [10] as no precise treatments such as hormone and targeted agents exist beyond non-specific anticancer therapeutics.

Due to such treatment limitations, a new combination of regimens using conventional anti-cancer drugs for TNBC is necessary.

Eribulin mesylate (eribulin) is a non-taxane, synthetic macrocyclic ketone analog of the marine natural product halichondrin B [11]. It is currently used to treat advanced and metastatic breast cancer. Clinical trial results have suggested that eribulin treatment can enhance the general survival of advanced breast cancer patients pretreated with anthracycline and taxane [12].

Cisplatin is a very effective chemotherapeutic agent used for therapy in cancers such as ovarian, cervical, bladder, testicular, lung, and breast cancers [13]. Cisplatin is a platinum drug that alkylates DNA by forming platinum-DNA adducts, causing DNA damage, G1/S cell cycle arrest, and apoptosis [14,15]. As a single agent, cisplatin is a standard chemotherapeutic drug, especially for TNBC [16]. Compared with other subtypes, TNBC is known to be more sensitive to platinum [17,18,19,20,21]. The proportion of BRCA1 mutations among TNBC patients was reported to be 9–100% (mean, 35%), and the proportion of BRCA2 mutations among TNBC patients was 2–12% (mean, 8%) [22]. *BRCA1 gene* products are involved in the process of homologously recombining DNA recovery pathways, particularly double-stranded disconnections [23]. TNBC preclinical models have a range of DNA repair disorders [24]. Therefore, DNA impairments caused by agents such as platinum drugs are assumed to be more effective for TNBC.

A previous clinical study demonstrated that combining eribulin and cisplatin had the potential to increase anticancer activity [25]. A recent study reported increased anticancer activity in HCC1806 and MX-1 TNBC cells with the compounds eribulin and platinum-based cisplatin and carboplatin [26].

Mitogen-activated protein kinase (MAPK) is strongly related to cancer and as the most studied signaling system in animal cells, it has been shown to affect cell growth, apoptosis, and cell division by transmitting external signals internally. The MAPK signaling pathway is triggered in TNBC. Three mammalian subfamilies of MAPKs have been revealed: extracellular signal-regulated kinase 1/2 (ERK1/2), p38 kinase, and c-Jun N-terminal kinase 1/2 (JNK1/2). ERK1 and ERK2 are well-characterized MAPKs that are activated with growth stimulation [27].

The objective of the present study was to investigate the function and possible mechanism of ERK1/2 activation in the cisplatin-induced cytotoxicity of TNBC by providing additional treatment with eribulin.

## 2. Materials and Methods

### 2.1. Reagents

Cisplatin was supplied by JW Pharmaceutical Corporation (JW Pharmaceutical CORPORATION, Seoul, Korea). Eribulin mesylate (1 mg/vial) was gifted by Eisai Co., Ltd. (Eisai Co., Ltd, Tokyo, Japan). PD98059 was acquired from Sigma Aldrich (Sigma Aldrich, Darmstadt, Germany).

### 2.2. Cell Lines and Culture

HCC38 (ATCC CRL-2314^TM^; ER-, PR-, HER2-), MDA-MB-231 (ATCC HTB-26^TM^; ER-, PR-, HER2-), and SKBR3 (ATCC HTB-30^TM^; HER2+) cells were acquired from the American Type Culture Collection (ATCC) (Manassas, VA, USA). RPMI-1640, Dulbecco’s modified Eagles Medium (DMEM), and fetal bovine serum were procured from HyClone (Logan, UT, USA). McCoy’s 5A was purchased from GIBCO (Waltham, MA, USA). The cells were grown in RPMI-1640 (HCC38), DMEM (MDA-MB-231), or McCoy’s 5A (SKBR3) media supplemented with 10% fetal bovine serum at a temperature of 37 °C in a humidified incubator with 5% CO_2_ and 95% air.

### 2.3. Cell Viability Assay

The cells were seeded into 96-well plates at a density of 5 × 10^3^ cells/mL and allowed to attach overnignt. Cisplatin and eribulin were supplemented to the media at varying concentrations. Following treatment, cell viability was assessed with a Cell Counting Kit-8 (CCK-8) (Dojindo Laboratories, Kumamoto, Japan). A CCK-8 solution (10 µL) was supplemented and incubated for three hours at 37 °C. Quantification was carried out using a precision microplate reader (Molecular Devices, LLC., San Jose, CA, USA).

### 2.4. Combination Index

After screening for the inhibitory effects of eribulin and cisplatin individually and in combination, we used the Chou-Talalay CI method with the CompuSyn software program(Combosyn Inc., Paramus, NJ, USA) to calculate the combination index (CI) [28]. The CI value is a quantitative determination of cisplatin and eribulin interactions, and CI values of <1, 1, and >1 indicate synergism, an additive effect, and antagonism, respectively.

### 2.5. Caspase-3/CPP32 Colorimetric Assay

Apoptosis was analyzed using the Caspase-3/CPP32 Colorimetric Assay Kit (BioVision, Mountain View, CA, USA), which calculates caspase activity by recognizing the DEVD sequence. The cells (1 × 10^6^ cells/mL) were treated with cisplatin, eribulin, and a combination of cisplatin and eribulin for 24 h. The cell lysates were prepared in chilled Cell Lysis Buffer according to the manufacturer’s instructions. Protein (100 µg) diluted to 50 µL with Cell Lysis Buffer was mixed with 50 µL of 2X Reaction Buffer (with 10 mM dithiothreitol) and 5 µL of DEVD-*p*NA substrate (200 µM final concentration). After incubation at a temperature of 37 °C for two hours, the absorbance was measured at 405 nm using a microplate reader.

### 2.6. Western Blot Analysis

The cells were lysed with a lysis buffer (1% TritonX-100, 10 mM Tris, pH 7.4, 1 mM ethylenediaminetetraacetic acid, 150 mM sodium chloride, 0.5% NP-40, 1 mM phosphatase inhibitor, 1 mM dithiothreitol, and 1 mM phenylmethylsulfonyl fluoride) and set on ice for one hour. A Pierce BCA Protein Assay Kit (Thermo Fisher Scientific Inc., Rockford, IL, USA) was used to measure the protein concentration of each sample. Equivalent quantities of protein (50 µg) were separated by 10% sodium dodecyl sulfate-polyacrylamide gel electrophoresis and transferred onto polyvinylidene difluoride membranes. The membranes were blocked with 5% skim milk in phosphate-buffered saline with 0.05% Tween-20 (PBST) for one hour at 25 °C to prevent nonspecific antibody binding. They were then incubated overnight with the appropriate primary antibodies (1:1000) at a temperature of 4 °C. After washing with PBST, the membranes were incubated for two hours with anti-rabbit horseradish peroxidase-conjugated IgG (1:3000) (Cell Signaling Technology, Inc., Danvers, MA, USA) at room temperature. Then, the membranes were visualized with an enhanced chemiluminescence (ECL) system using Super Signal West Pico Chemiluminescent Substrate (Thermo Fisher Scientific Inc, Rockford, IL, USA.). The primary antibodies to phospho-p44/p42 MAPK (ERK1/2) (Thr202/Tyr204), poly-(ADP-ribose) polymerase (PARP), and β-actin were procured from Cell Signaling Technology, Inc. The primary antibody to ERK was supplied by Santa Cruz Biotechnology, Inc. (Dallas, TX, USA). The density of the Western blot bands was quantified with ImageJ software and normalized to loading controls. ImageJ was downloaded from the National Institutes of Health website (http:rsbweb.nih.gov/ij/download.html, accessed on 17 June 2021).

### 2.7. Statistical Analysis

The statistical significance of the differences was analyzed using IBM^®^ SPSS^®^ Statistics version 24.0 (IBM Corp., Armonk, NY, USA). One-way ANOVA was used to calculate the significance between different groups. Values are given as the means ± standard deviation of three independent experiments. Statistical significance was expressed by *p* < 0.05 and *p* < 0.01.

## 3. Results

### 3.1. Growth Inhibitory Effects of Single-Agent Eribulin or Cisplatin in HCC38, MDA-MB-231, and SKBR3 Breast Cancer Cell Lines

The HCC38 and MDA-MB-231 cells were treated with cisplatin or eribulin for 24 h to investigate the growth inhibitory effects of these agents on TNBC cell lines. A Cell Counting Kit-8 (CCK8) was used to measure cell viability. The SKBR3 (Her2+) cells were used as negative control cells (non-TNBC). Cisplatin (3, 15, 30, and 60 µM) inhibited the development of HCC38, MDA-MB-231, and SKBR3 cells in a dose-dependent manner (Figure 1A). The IC_50_ values for cisplatin were acquired (37.58 ± 0.08 µM in HCC38 cells, 40.34 ± 0.21 µM in MDA-MB-231 cells, and 22.87 ± 0.16 µM in SKBR3 cells) from the dose-effect curves. A single treatment with eribulin (12, 30, 60, and 120 µM) for 24 h showed no significant growth inhibition in any of the three cell lines (Figure 1B). However, eribulin treatment for 48 h and 72 h was cytotoxic in each of the HCC38, MDA-MB-231, and SKBR3 cell lines, as shown by the decreased cell viability (data not shown). The IC_50_ values for eribulin exceeded 200 µM in all three breast cancer cell lines.

### 3.2. Apoptotic Activity of Single-Agent Eribulin or Cisplatin in HCC38, MDA-MB-231, and SKBR3 Breast Cancer Cell Lines

To examine the ability of cisplatin or eribulin to cause apoptosis in breast cancer cell lines, the cells were treated with varying concentrations of cisplatin or eribulin for 24 h. Western blot analysis was then performed to measure the cleavage of PARP, a biochemical feature of apoptosis. Cisplatin treatment significantly induced the cleavage of PARP at concentrations of 30 µM and 60 µM (Figure 2A). However, eribulin treatment for 24 h did not cause apoptotic cell death (Figure 2B), although an extended eribulin treatment time of 48 h and 72 h showed apoptotic effects (data not shown).

### 3.3. Effect of Cisplatin and Eribulin Combination on HCC38, MDA-MB-231, and SKBR3 Cells

Treatment of cisplatin (1.5, 3, 15, and 30 µM) in combination with eribulin (60 µM) for 24 h significantly decreased the viability of the HCC38 and MDA-MB-231 TNBC cells, but not that of the SKBR3 cells (Figure 3).

The Chou-Talalay method was used to calculate the CI (combination index) values to evaluate the combined synergistic effect of cisplatin and eribulin (24). The CI value for the combination of cisplatin (30 µM) and eribulin (60 µM) was less than one, indicating a synergistic interaction between cisplatin and eribulin in suppressing the growth of the TNBC cell lines (HCC38 and MDA-MB-231) (Table 1). We found synergistic action of the combined treatment of TNBC cells with cisplatin and eribulin.

### 3.4. Combination of Cisplatin and Eribulin Significantly Enhances the Apoptosis of HCC38 and MDA-MB-231 TNBC Cells

As the cisplatin and eribulin combination significantly decreased the viability of TNBC cells, we investigated whether the enhanced cytotoxic effect of the combination was due to cell apoptosis. To examine apoptotic activity in the cells, we measured PARP cleavage by Western blot analysis and evaluated caspase-3 activity levels with a Caspase-3/CPP32 Colorimetric Assay Kit. All three cell lines were treated with cisplatin, eribulin, or a combination of cisplatin and eribulin for 24 h. PARP protein cleavage was significantly increased in the HCC38 and MDA-MB 231 cells after treatment with the cisplatin (15 or 30 µM) and eribulin (60 µM) combination for 24 h. No synergistic effect of the combination on PARP cleavage was observed in SKBR3 cells (Figure 4A). Caspase-3 activity was also increased by 1.8-fold and 1.43-fold in the HCC38 and MDA-MB-231 cells, respectively, following the combined treatment compared with cisplatin-only (Figure 4B). Such results showed that the cisplatin and eribulin combination could significantly increase the apoptosis of TNBC cells by increasing PARP cleavage and caspase-3 activation.

### 3.5. Combination of Cisplatin and Eribulin Synergistically Activates ERK1/2 in TNBC Cells

The cells were treated with cisplatin (3, 15, 30, and 60 µM) or eribulin (12 and 60 µM) for 24 h. The phosphorylation level of ERK1/2 was then determined by Western blot analysis with antibodies against phospho-ERK. Cisplatin treatment resulted in potent ERK activation in the HCC38, MDA-MB-231, and SKBR3 cells (Figure 5A). The activation was seen in the HCC38 and MDA-MB-231 cells after treatment with cisplatin (30 and 60 µM), which was similar to the pattern shown in Figure 1A and Figure 2A. In the SKBR3 cells, ERK activation was sustained after treatment with 30 µM cisplatin, and apparent with 60 µM cisplatin (Figure 4A). No significant ERK phosphorylation level was found in any of the three cell lines after 24 h of eribulin treatment (Figure 5B). The effect of the combination treatment on ERK activation was then examined. The combined treatment of cisplatin (30 µM) and eribulin (60 µM) for 24 h significantly increased ERK phosphorylation in the HCC38 and MDA-MB-231 cells. However, the treatment decreased ERK phosphorylation in the SKBR3 cells (Figure 5C). The results suggest that eribulin increased cisplatin-induced cell death via synergistic ERK activation in TNBC cells.

### 3.6. Effect of ERK1/2 Inhibitor PD98059 on the Sensitivity of TNBC to the Cisplatin and Eribulin Combination

The possible role of ERK1/2 in the synergistic effect of the cisplatin and eribulin combination in inducing TNBC cell death was evaluated using the pharmacological ERK1/2 inhibitor PD98059. In all of the cell lines tested, a concentration of 25 µM PD98059 effectively blocked the basal phosphorylation of ERK1/2. Figure 6A presents ERK1/2 phosphorylation in cells treated with the cisplatin and eribulin combination (lane nos. 2, 6, and 10) and additional PD98059 (lane nos. 3, 7, and 11). PD98059 significantly reduced the phosphorylation level of ERK1/2 in TNBC cells treated with cisplatin and eribulin. The upregulation of ERK1/2 activity after treatment with the combination of cisplatin and eribulin was accompanied by a decrease in PARP cleavage and cytotoxicity in the cells. PD98059 effectively decreased PARP cleavage (Figure 6B) and increased cell viability in TNBC cells after combined treatment for 24 h (Figure 6C). However, PD98059 did not significantly influence SKBR3 cell viability, ERK1/2 phosphorylation, or PARP cleavage (Figure 6). The results suggest that eribulin was involved in the cisplatin-induced death of TNBC cells through ERK1/2 activation.

## 4. Discussion

TNBC patients do not show therapeutic responses to hormone or targeted treatments due to the loss of estrogen receptors (ERs), progesterone receptors (PRs), and human epidermal growth factor 2 (HER2) receptors [29]. The DNA repair complex (platinum compounds and taxanes), cell proliferation (anthracycline), p53 (taxanes), immune checkpoints, PARP, etc., can all be targeted as strategies for the therapeutic management of TNBC [30]. However, effective therapeutic targets and treatment for TNBC are still limited. Many patients with TNBC become refractory to the treatment. Thus, novel therapeutic strategies to treat TNBC or enhance the survival of TNBC patients are urgently needed.

Single-agent therapies such as anticancer and targeted agents showed positive effects on preclinical models, but they have been unsuccessful in delivering encouraging results for managing aggressive TNBC in clinical trials because of heterogeneous clinical behavior and the acquisition of drug resistance [30].

Currently, about 80% of clinical trials use drug combinations to explore new therapeutic strategies for TNBC management [30]. A recent study showed that the combination of cisplatin and metformin could enhance the anticancer effect by downregulating RAD51 in TNBC [31]. Combined treatment of the PI3K/mTOR inhibitor NVP-BEZ235 with cisplatin completely reversed cisplatin resistance in TNBC [32]. Another study reported that the combination of the immunomodulatory drug lenalidomide and cisplatin could improve anticancer efficacy in TNBC [33]. Several studies have demonstrated the combination of eribulin and cisplatin to be effective in treating ovarian cancer (A2780) and TNBC (HCC1806 and MX-1) [26], as well as advanced solid tumors (breast, pancreatic, bladder, head and neck, and lung cancer) [25].

In this study, eribulin enhanced the anticancer activity of cisplatin in TNBC cells. The study data revealed that cisplatin alone decreased cell viability and promoted apoptotic activity in the breast cancer cell lines HCC38, MDA-MB-231, and SKBR3, whereas eribulin alone had no anticancer effect after treatment for 24 h. Moreover, the eribulin and cisplatin combination significantly inhibited cell growth and induced apoptotic activity in TNBC cells (HCC38 and MDA-MB-231). Upon discovering the important effects of the combined treatment of eribulin and cisplatin in TNBC cells, we investigated the mechanism of their actions. We discovered that the eribulin and cisplatin combination greatly increased levels of p-ERK1/2 proteins only in TNBC cells.

We examined whether cisplatin plus eribulin could increase the death of TNBC cells through an apoptotic mechanism compared with the use of cisplatin alone. One study reported that cisplatin at concentrations of 25 to 100 µM caused the apoptosis of MDA-MB-231 cells after 24 h of incubation [34]. We found that cisplatin concentrations ≥ 30 µM were significantly cytotoxic to HCC38 and MDA-MB-231 cells. Treatment with cisplatin at concentrations ≥ 15 µM for 24 h was also cytotoxic to SKBR3 cells. Adding 60 µM eribulin to 15 and 30 µM cisplatin greatly decreased TNBC cell viability. The same concentrations of eribulin and cisplatin resulted in significantly enhanced apoptosis after combined treatment for 24 h compared with single treatments of cisplatin or eribulin. We have yet to find an existing report on the cytotoxic or apoptotic effects of the combination of eribulin and cisplatin on TNBC cells.

In addition, the level of ERK1/2 phosphorylation was synergistically increased by the cisplatin and eribulin combination compared with single treatments of cisplatin or eribulin. Cisplatin was shown to activate ERK in several cancer cell types such as bladder cancer [35], renal cancer [36], and glioma [37]. Some studies reported that ERK activation was associated with increased cell survival and proliferation by cisplatin treatment [38,39,40]. In contrast, other studies demonstrated that ERK activation caused apoptosis after treatment with cisplatin [36,37,41]. Therefore, it is unclear whether ERK signaling is associated with apoptosis or cell death in cells treated with cisplatin. This study reports a significant increase in ERK1/2 activation in breast cancer cells after cisplatin treatment at concentrations ≥ 30 µM (HCC38 and MDA-MB-231) or 60 µM (SKBR3). Clinically, the combination of cisplatin and a microtubule-targeting drug such as eribulin has the potential to increase anticancer activity [25]. However, the underlying mechanisms involved in the role of ERK activation in eribulin-treated cells are not fully comprehended.

## 5. Conclusions

The preliminary data in this study indicated that eribulin in combination with cisplatin could be used in TNBC treatment and that ERK1/2 plays a crucial role in the synergistic effect between eribulin and cisplatin. The effects of combined treatment with eribulin and cisplatin can contribute to TNBC treatment options for patients with poor prognosis and therapeutic difficulties.

## Figures and Tables

**Figure 1 medicina-58-00547-f001:**
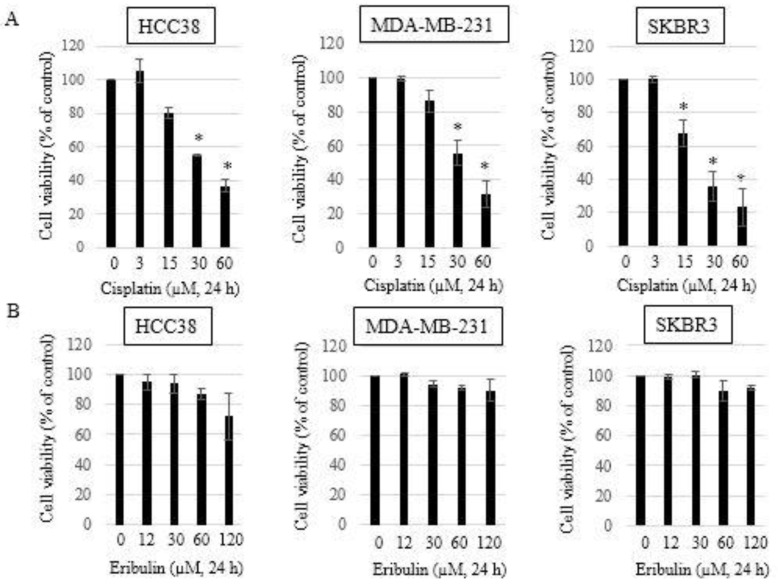
Effect of cisplatin and eribulin on the viability of HCC38, MDA-MB-231, and SKBR3 breast cancer cells. Cells were treated with cisplatin (3, 15, 30, and 60 µM) (**A**) or eribulin (12, 30, 60, and 120 µM) (**B**) for 24 h. Cell viability was measured using the Cell Counting Kit-8 assay. Each assay was performed in triplicate. Data are presented as the mean ± standard deviation; * *p* < 0.05 vs. untreated control.

**Figure 2 medicina-58-00547-f002:**
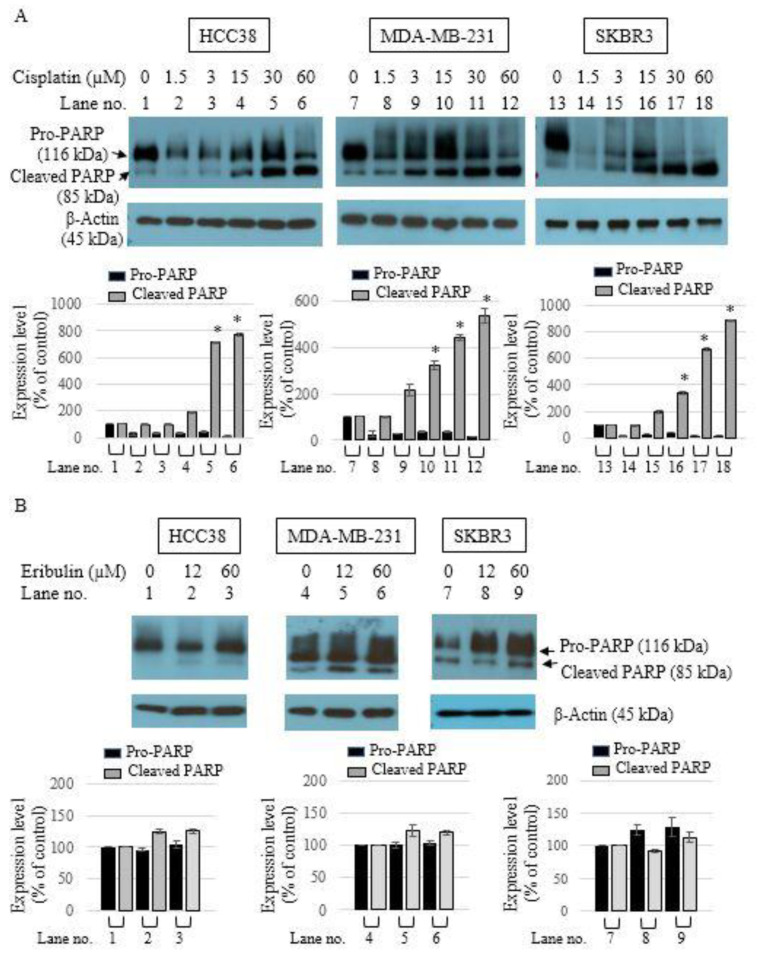
Apoptotic effect of cisplatin or eribulin on PARP cleavage in HCC38, MDA-MB-231, and SKBR3 breast cancer cells. Cells were treated with cisplatin (1.5, 3, 15, 30, and 60 µM) (**A**) or eribulin (12 and 60 µM) (**B**) for 24 h. Cleavage of the caspase substrate PARP in the presence of cisplatin or eribulin is shown. Cells were harvested after 24-h cisplatin or eribulin treatment. PARP cleavage was assessed by Western blot analysis using an anti-PARP antibody. β-Actin was used as the internal control. Data are presented as the mean ± standard deviation; * *p* < 0.05 vs. untreated control.

**Figure 3 medicina-58-00547-f003:**
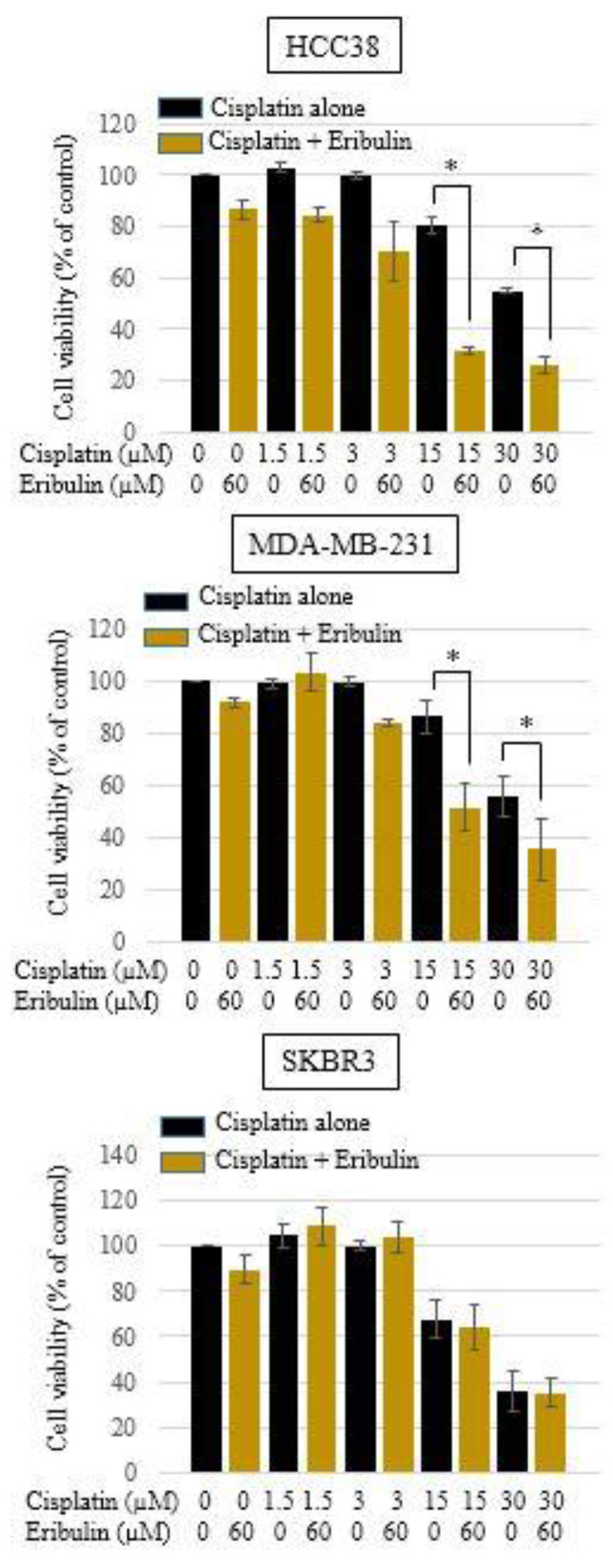
Combined effect of cisplatin and eribulin on HCC38, MDA-MB-231, and SKBR3 breast cancer cells. Cells were treated with cisplatin (1.5, 3, 15, and 30 µM) or a combination of cisplatin and eribulin (60 µM). After 24 h of exposure, cell viability was measured using a Cell Counting Kit-8 assay. Each assay was performed in triplicate. Data are presented as the mean ± standard deviation; * *p* < 0.05 vs. cisplatin-only treatment.

**Figure 4 medicina-58-00547-f004:**
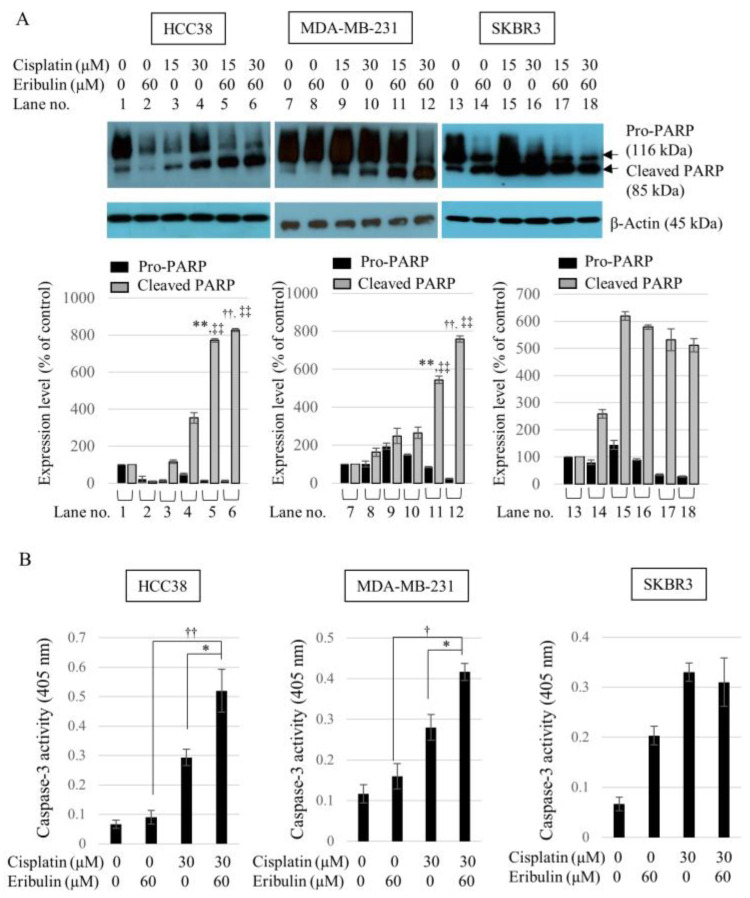
Eribulin enhances the apoptotic effect of cisplatin on TNBC cells. (**A**) HCC38, MDA-MB-231, and SKBR3 cells were exposed to cisplatin (15 and 30 µM) or eribulin (60 µM) individually or in combination for 24 h. Cell lysates were collected and subjected to Western blot analysis with the anti-PARP antibody. β-Actin was used as the internal control; ** *p* < 0.01 vs. cisplatin-only (15 µM) treatment in cleaved PARP, ^††^
*p* < 0.01 vs. cisplatin-only (30 µM) treatment in cleaved PARP, ^‡‡^
*p* < 0.01 vs. eribulin-only (60 µM) treatment in cleaved PARP. (B) Cells were treated with cisplatin alone, eribulin alone (60 µM), or a combination of cisplatin and eribulin for 24 h. The absorbance of pNA (p-nitroaniline) at 405 nm was measured. Results are presented as the mean ± standard deviation from three independent studies; * *p* < 0.05 vs. cisplatin-only (30 µM) treatment, ^†^
*p* < 0.05, ^††^
*p* < 0.01 vs. eribulin-only (60 µM) treatment.

**Figure 5 medicina-58-00547-f005:**
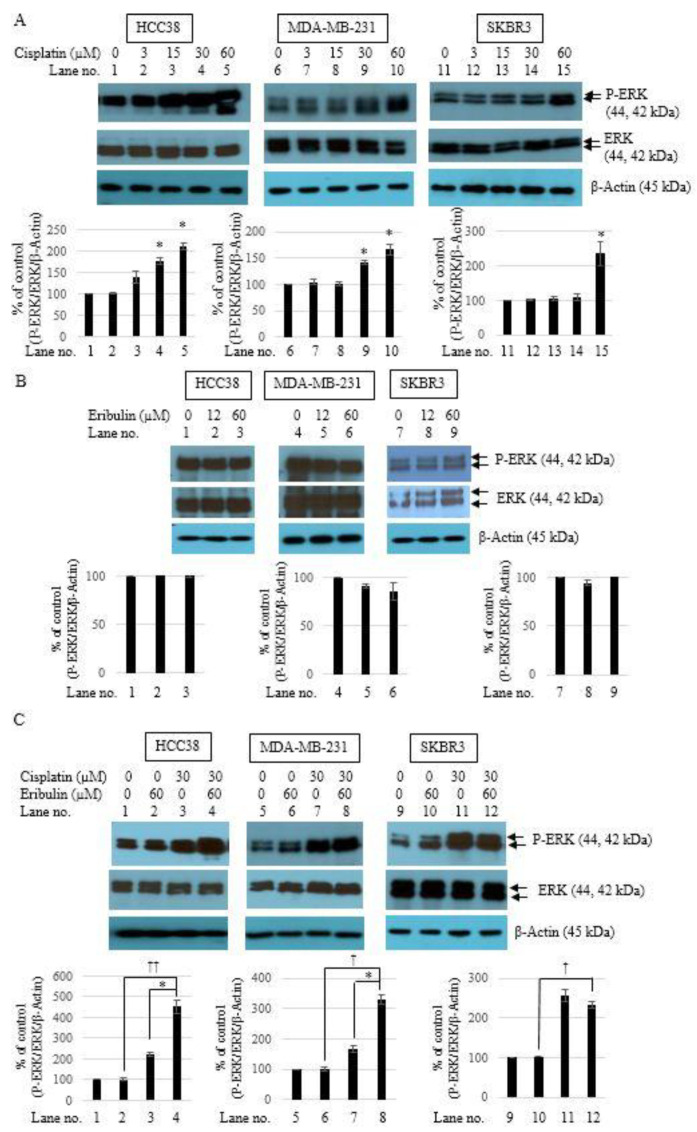
Eribulin and cisplatin in combination synergistically activated ERK activation in TNBC cells. HCC38, MDA-MB-231, and SKBR3 cells were exposed to cisplatin (3, 15, 30, and 60 µM) (**A**) or eribulin (12 and 60 µM) (**B**) for 24 h. Cell lysates were collected and subjected to Western blot analysis with anti-phospho-ERK1/2 and anti-ERK1/2 antibodies. β-Actin was used as the internal control. Data are presented as the mean ± standard deviation; * *p* < 0.05 vs. untreated control. (**C**) Cells were exposed to cisplatin (30 µM) or eribulin (60 µM) individually or in combination for 24 h. Levels of phosphorylated ERK1/2 and total-ERK1/2 were assessed by Western blot analysis. β-Actin was used as the internal control; * *p* < 0.05 vs. cisplatin-only (30 µM) treatment, ^†^
*p* < 0.05, ^††^
*p* < 0.01 vs. eribulin-only (60 µM) treatment.

**Figure 6 medicina-58-00547-f006:**
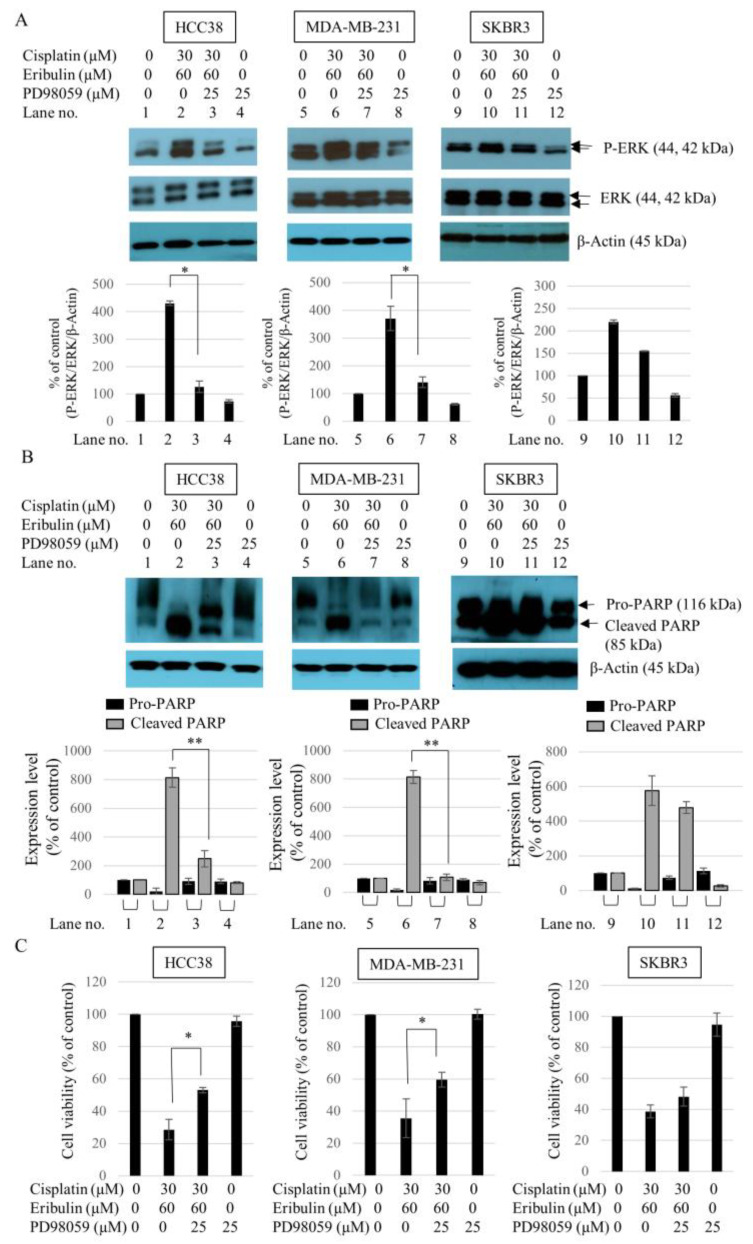
Phosphorylation of ERK1/2 in HCC38, MDA-MB-231, and SKBR3 cells treated with PD98059 and the combination of cisplatin and eribulin. (**A**) Cells were first treated with 25 µM PD98059 for 2 h and then cultured with cisplatin (30 µM) and eribulin (60 µM) for 24 h. Total ERK1/2 and phospho-ERK1/2 levels were analyzed by Western blot analysis. β-Actin was used as the internal control; * *p* < 0.05 vs. cisplatin (30 µM) + eribulin (60 µM). (**B**) Cell lysates were collected and subjected to Western blot analysis with the anti-PARP antibody. β-Actin was used as the internal control; ** *p* < 0.01 vs. cisplatin (30 µM) + eribulin (60 µM) treatment in cleaved PARP. (**C**) After 24 h exposure, cell viability was measured using a Cell Counting Kit-8 assay. Each assay was performed in triplicate. Data are presented as the mean ± standard deviation; * *p* < 0.05 vs. cisplatin (30 µM) + eribulin (60 µM).

**Table 1 medicina-58-00547-t001:** Combination index (CI) values for cisplatin and eribulin combination for breast cancer cell lines.

Cell Lines	Cispatin (µM)	Eribulin (µM)	CI	DRI (Cisplatin)	DRI (Eribulin)
HCC38	3	12	0.45	13.91	2.62
	15	30	0.40	8.05	3.56
	30	60	0.49	4.06	24.65
	60	120	0.45	2.37	34.13
MDA-MB-231	3	12	8.15	1.0	0.14
	15	30	0.58	4.8	2.67
	30	60	0.53	3.0	12.19
	60	120	0.44	2.5	23.29
SKBR3	3	12	4.68	ND	ND
	15	30	5.56	ND	ND
	30	60	1.27	ND	ND
	60	120	1.07	ND	ND

Abbreviations: CI, combination index; DRI, dose reduction index; ND, not determined. The combination was synergistic at CI < 1, additive at CI = 1, and antagonistic at CI > 1.

## Data Availability

Not applicable.

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
