# Peer review of "Eribulin Mesylate Improves Cisplatin-Induced Cytotoxicity of Triple-Negative Breast Cancer by Extracellular Signal-Regulated Kinase 1/2 Activation"

_medicina, 2022, doi:10.3390/medicina58040547_

Round 1

Reviewer 1 Report

Authors submitted "Eribulin Mesylate Improves Cisplatin-Induced Cytotoxicity of Triple-Negative Breast Cancer by Extracellular Signal-Regulated Kinase 1/2 Activation". In this article, the combined effects of  Eribulin Mesylate and cisplatin on TNBC were evaluated. Therefore, these findings will contribute to new insights into the literature. However, some major points should be addressed.

  1. The use of the English language is rather poor, leading to unresolvable phrasing in places.
  2.  In the introduction section, the authors should re-written the TNBC part. The authors mention the aggressive features of TNBC  rather than BRCA mutation status.
  3. How do you select eribulin concentrations because different concentrations have been used in the literature? 
  4. In this article, the main aim is the therapeutic role of combination drugs in TNBC. Why author select SKBR3 cells ?
  5. How do the authors select combination concentrations and exposure time for each cisplatin or eribulin and their combination?
  6. The response rate of each cell to cisplatin alone or in combination was different. Please discuss the different response rates of the cell line in terms of genetic background.
  7. Table 1 should be revised. The authors add each CI value for each combination. Additionally, please add DRI value of each combination.
  8. In Fig 4 and Fig 6, the quality of western blot bands is so low. Please revise the results. 
  9. The authors did not use a control cell line for especially combination studies. Therefore, at least the cytotoxic effects of combination doses should be added for control cell line.
  10. Authors should add any quantitative results (Annexin V, caspase et al.) for apoptotic effects. Authors only detect PARP and cPARP levels. Where is the results of the Caspase-3/CPP32 Colorimetric Assay Kit in Fig 4 A and B?

Author Response

Point 1: The use of the English language is rather poor, leading to unresolvable phrasing in places

Response 1: The previous version underwent English editing before submission. To improve it further, additional editing services were obtained. I attched new certificate of editing.

Point 2: In the introduction section, the authors should re-written the TNBC part. The authors mention the aggressive features of TNBC rather than BRCA mutation status.

Response 2: BRCA mutation was mentioned as a characteristic of TNBC. To eliminate any ambiguity, the sentence was divided and rearranged as follows: Also, BRCA1 mutation carriers [7, 8] are more likely to have TNBC. And TNBC is associated with more aggressive tendencies compared to other breast cancer subtypes.

Point 3: How do you select eribulin concentrations because different concentrations have been used in the literature? 

Response 3: The concentration of eribulin for injection was 1 vial = 1 mg/2 mL = 0.5 mg/mL = 0.6 mM, and the experimental concentrations were 0, 12, 30, 60, and 120 mM. Cytotoxicity by eribulin was expected at the time of the first experiment. However, when treated for 24 hours, no cytotoxicity was observed from low concentrations (nM) to a high concentration (120 mM). When the treatment time was increased to 48 hours or 72 hours, cytotoxicity was observed (data not included). A concentration of 60 mM, which is considered to be a fairly high concentration, was selected and combined with cisplatin, and significant cytotoxicity was observed, so this concentration was selected.

Point 4: In this article, the main aim is the therapeutic role of combination drugs in TNBC. Why author select SKBR3 cells ?

Response 4: In the initial experiment, only two types of TNBC cell lines, HC38 and MDA-MB-231, were used. However, to investigate the effect on non-TNBC cell lines, SKBR3, a non-TNBC cell line stored in our laboratory, was selected and tested.

Point 5: How do the authors select combination concentrations and exposure time for each cisplatin or eribulin and their combination?

Response 5: For the cisplatin concentration, a concentration showing about 50% cell viability in three cell lines was selected (Figure 1), and the maximum concentration for eribulin was selected using 1 vial (0.6 mM) (please see the rationale in response 3 above). Although only the results of treatment for 24 hours were shown in this study, the effect of combined treatment continued up to 48 hours and 72 hours. Thus, the results of long-term treatment (48 and 72 hours) showed a significantly different mechanism from that of short-term (24 hours). In this study, only the results of treatment within 24 hours were presented. Because the experimental results seen within 24 hours were very characteristic, the research was judged to be worthwhile and the results were reported. Follow-up studies are planned.

Point 6: The response rate of each cell to cisplatin alone or in combination was different. Please discuss the different response rates of the cell line in terms of genetic background.

Response 6: The breast cancer cell lines used in this experiment were (1) SKBR3 =HER2 +, (2) HCC38 = HER2(-), p53+, ER(-), PR(-), and (3) MDA-MB-231 = EGF(+), TGFa (+). However, it was not possible to evaluate the response rate for each cell line. Experimental results were judged by classifying only TNBC (HCC39 and MDA-MB-231) and non-TNBC (SKBR3). I think it is difficult to evaluate the kinetics of a cell line related to the genetic background.

Point 7: Table 1 should be revised. The authors add each CI value for each combination. Additionally, please add DRI value of each combination.

Response 7: The table was revised as follows.(Attached as a file)

Point 8: In Fig 4 and Fig 6, the quality of western blot bands is so low. Please revise the results. 

Response 8: The resolution was revised.(Attached as a file)

Point 9: The authors did not use a control cell line for especially combination studies. Therefore, at least the cytotoxic effects of combination doses should be added for control cell line.

Response 9: In this experiment, the non-TNBC cell line, SKBR3, was added as the control cell line.

Point 10: Authors should add any quantitative results (Annexin V, caspase et al.) for apoptotic effects. Authors only detect PARP and cPARP levels. Where is the results of the Caspase-3/CPP32 Colorimetric Assay Kit in Fig 4 A and B?

Response 10: In this experiment, only PARP cleavage, which is the intermediate process of apoptosis, was measured, and annexin V staining, which visualizes total apoptosis, was not performed. In the next study, annexin V will be included to improve the completeness of the study.

Reviewer 2 Report

The paper by Ko et al. sought to investigate the effects of Eribulin in triple-negative breast cancer lineages and found that it may synergize with cisplatin in inducing anti-proliferative and apoptotic response specifically in cell lineages with triple-negative profile, but not in a HER2 overexpressing breast cancer cells. Also, by investigating ERK1/2 phosphorylation and by using a specific ERK1/2 inhibitor, authors implicate this pathway in the observed phenomenon. Although the study report somewhat preliminary in vitro results, which must be validated in in vivo to suggest eribulin as a potential adjuvant treatment for TNBC, the reported results are promisor, and consistent and greatly describe an interesting phenomenon, providing mechanistical insights into it. Therefore, I only have minor points described below to improve the paper clarity and quality.

1) In the abstract please reformulate the phrase "Administration of PD98059 significantly changed ERK1/2 activation, decreased PARP cleavage, and increased the viability of TNBC cells." to inform that PD98059 is a ERK1/2 inhibitor and that the increase in TNBC cells viability was observed in the context of treatment with cisplatin plus eribulin.

2) In line 50 of introduction, the phrase "At least one third of BRCA1 mutant carriers suffer from TNBC." Seems misleading, since it gives the wrong idea that 33% of all BRCA1 mutant carriers in the general population have TNBC. It should be corrected to make clear that this proportion occurs among breast cancer patients.

3) Lines 143-145: The phrase "However, eribulin treatment for 48 hours and 72 hours displayed cytotoxicity in various breast cancer cells (data not shown)." is too vague. Please specify at least which breast cancer cells were tested and briefly describe how the cytotoxic effect was defined. Ideally, these results could be shown as supplemental data and the information could be supplemented by citations to relevant literature.

4) Lines 247-249: The phrase "Downregulated ERK1/2 activity after treatment with the combination of cisplatin and eribulin was accompanied by PARP cleavage and cytotoxicity in cells." is confusing, since according to figure 6C authors have observed an increment in ERK1/2 activity following treatment with cisplatin and eribulin, and not a decrease, as written. Please clarify.

5) Lines 267-269: The phrase "Targeting DNA repair complex (platinum compounds and taxanes), cell proliferation (anthracyline), p53 (taxanes), and immune checkpoint inhibitor (ICI), PARP inhibitor, etc." is confusing, since immune checkpoint inhibitors and PARP inhibitors are classes of treatments, and not targets as the other examples cited. Correcting to "...immune checkpoints, PARP, etc." would be fine.

6) Line 289: The phrase "The study data revealed that cisplatin alone inhibited cell viability and apoptotic activity" is also confusing since it gives the wrong idea that cisplatin inhibited apoptotic activity, when it instead promoted it.

Author Response

Point 1: In the abstract please reformulate the phrase "Administration of PD98059 significantly changed ERK1/2 activation, decreased PARP cleavage, and increased the viability of TNBC cells." to inform that PD98059 is a ERK1/2 inhibitor and that the increase in TNBC cells viability was observed in the context of treatment with cisplatin plus eribulin.

Response 1: Correct as follows : Administration of ERK1/2 inhibitor PD98059 increased the viability of TNBC cells treated with cisplatin plus eribulin.

Point 2: In line 50 of introduction, the phrase "At least one third of BRCA1 mutant carriers suffer from TNBC." Seems misleading, since it gives the wrong idea that 33% of all BRCA1 mutant carriers in the general population have TNBC. It should be corrected to make clear that this proportion occurs among breast cancer patients

Response 2: It was incorrect to explain the relationship between BRCA mutations and TNBC with ambiguous numbers, so more specific statistics were presented. Correct as follows : The proportion of BRCA1 mutations among TNBC patients was reported to be 9–100% (mean, 35%), and the proportion of BRCA2 mutations among TNBC patients was 2–12% (mean, 8%)

Point 3: Lines 143-145: The phrase "However, eribulin treatment for 48 hours and 72 hours displayed cytotoxicity in various breast cancer cells (data not shown)." is too vague. Please specify at least which breast cancer cells were tested and briefly describe how the cytotoxic effect was defined. Ideally, these results could be shown as supplemental data and the information could be supplemented by citations to relevant literature. 

Response 3: When eribulin was used alone, there was no cytotoxic effect within 24 hours. When tested in the same cell line, cytotoxicity appeared after 24 hours. The cytotoxic effect was derived by evaluating cell viability. Correct as follows : However, eribulin treatment for 48 hours and 72 hours was cytotoxic in each of the HCC38, MDA-MB-231, and SKBR3 cell lines, as shown by decreased cell viability (data not shown).

Point 4: Lines 247-249: The phrase "Downregulated ERK1/2 activity after treatment with the combination of cisplatin and eribulin was accompanied by PARP cleavage and cytotoxicity in cells." is confusing, since according to figure 6C authors have observed an increment in ERK1/2 activity following treatment with cisplatin and eribulin, and not a decrease, as written. Please clarify.

Response 4: ERK1/2 decreased when PD98059, an ERK1/2 inhibitor, was used after combined treatment with cisplatin and eribulin (Figure 6A). In the TNBC cell lines HCC38 and MDA-MB-231, PARP-cleavage was reduced (Figure 6B) and cell viability was increased (Figure 6C). It seems there is confusion in the content, so content on decreases in PARP cleavage was added. Correct as follows : The upregulation of ERK1/2 activity after treatment with the combination of cisplatin and eribulin was accompanied by a decrease in PARP cleavage and cytotoxicity in the cells.

Point 5: Lines 267-269: The phrase "Targeting DNA repair complex (platinum compounds and taxanes), cell proliferation (anthracyline), p53 (taxanes), and immune checkpoint inhibitor (ICI), PARP inhibitor, etc." is confusing, since immune checkpoint inhibitors and PARP inhibitors are classes of treatments, and not targets as the other examples cited. Correcting to "...immune checkpoints, PARP, etc." would be fine.

Response 5: Correct as follows : Targeting the DNA repair complex (platinum compounds and taxanes), cell proliferation (anthracyline), p53 (taxanes), immune checkpoints, PARP, etc. are strategies for the therapeutic management of TNBC.

Point 6: Line 289: The phrase "The study data revealed that cisplatin alone inhibited cell viability and apoptotic activity" is also confusing since it gives the wrong idea that cisplatin inhibited apoptotic activity, when it instead promoted it.

Response 6: Correct as follows : The study data revealed that cisplatin alone decreased cell viability and promoted apoptotic activity in the breast cancer cell lines HCC-38, MDA-MB-231, and SKBR3, whereas eribulin alone had no anticancer effect after treatment for 24 hours.

Round 2

Reviewer 1 Report

Author revised the manuscript according to my comments. Now it is published.